

# Adaptive classification of artistic images using multi-scale convolutional neural networks

Jin Xiang[1], Yi Yang[2] and Junwei Bai[1]

[1] School of Art and Design, Wuhan Polytechnic University, Wuhan, China
[2] School of Industrial Design, Hubei Institute of Fine Arts, Hubei, Wuhan, China

## ABSTRACT

The current art image classification methods have low recall and accuracy rate issues . To improve the classification performance of art images, a new adaptive classification method is designed employing multi-scale convolutional neural networks (CNNs). Firstly, the multi-scale Retinex algorithm with color recovery is used to complete the enhancement processing of art images. Then the extreme pixel ratio is utilized to evaluate the image quality and obtain the art image that can be analyzed. Afterward, edge detection technology is implemented to extract the key features in the image and use them as initial values of the item to be trained in the classification model. Finally, a multi-scale convolutional neural network (CNN) is constructed by using extended convolutions, and the characteristics of each level network are set. The decision fusion method based on maximum output probability is employed to calculate different subclassifies' probabilities and determine the final category of an input image to realize the art image adaptive classification. The experimental results show that the proposed method can effectively improve the recall rate and precision rate of art images and obtain reliable image classification results.

## INTRODUCTION

Improving people's quality of life and appreciation level has led many to engage in artistic image creation and collection efforts. The current classification management is mainly achieved manually by professionals when faced with art images, and the human and financial costs are relatively high. Therefore, it is of great significance to study how to classify all kinds of art images efficiently and accurately, to help users screen images that meet their needs better (*Wang et al., 2023*). The development of new information technology such as big data and artificial intelligence has also made artistic creation more diverse and artistic styles more abundant (*Liu, 2022*; *Li, 2021*).

Recent developments have brought remarkable breakthroughs dependent upon content-oriented image analysis and retrieval research. However, considering the unique stroke style of artistic works, conventional classification algorithms find it difficult to classify them directly (*Alzahem et al., 2023a*). An image classification framework was designed

Corresponding author
Jin Xiang, 22113037@whpu.edu.cn

based on contrast self-supervised learning and more effective agent tasks were selected to improve the robustness of the model that uses a proposed targeted loss function to improve the performance of image classification (*Zhao et al., 2022*). However, this method can be used for the basic classification of artistic images, and the classification effect is general. An image classification method was proposed based on a multi-branch bottleneck structure (*Su, Wang & Zhang, 2023*). This method increases the feature diversity by using multi-branch structures in the backbone of the residual structure, reduces the number of model parameters by utilizing the depth-separable convolution of variants, and then increases the nonlinear expression capability of the network by implementing Mish activation functions, which improves the classification accuracy while effectively reducing the model volume. However, it is found that the accuracy of the classification results is low in practical implementations (*Tang et al., 2023a*). An adaptive image classification method was proposed based on a task-associated feature decoupling network. To learn the consistency of the attention between features and original features under interdomain style mixing interference, the network can extract the feature weights related to downstream tasks, and further use weight difference to obtain task-independent feature weights. Then, the orthogonal function constraint is employed to deduce the task-associated and irrelevant features, and the feature decoupling is realized. Afterward, the task features are designed to resolve the coupling layer, reduce the confusion between the paired features and the unique features of the domain, and optimize the accuracy of the classification model. However, after this method is applied to art image processing, it is found that the recall rate of the obtained classification results is low, and it needs to be optimized.

A convolutional network is a multi-layer perceptron specially designed to recognize two-dimensional shapes. This network structure is invariant to translation, scaling, and other forms of deformation (*Ko & Chung, 2023*; *Tang et al., 2023b*). This advantage can reduce the complexity of feature extraction and reconstruction. Therefore, this study selects a multi-scale convolutional neural network (CNN) to process the current art images and proposes a new adaptive classification method for art images.

Traditional Indian art forms that enjoy widespread popularity across the world, called the Madhubani style, were investigated by Transfer learning based-CNN to identify five distinct groups of Madhubani art, namely, Bharni, Godna, Kachni, Kohbar, and Tantrik (*Liu et al., 2021*). The main purpose was to go a long way in preserving precious art heritage and fostering its rapid growth in the world market. The correct identification of artistic styles or art movements of paintings is pivotal when large artistic databases are used to index, and find painters' authentication. Thus, feature extraction is a very critical stage in conducting classification efforts. The article aims to derive significant features, namely, two unique colors and one texture feature by employing CNN (*Cetinic, Lipic & Grgic, 2018*). CNN open new research perspectives for the increased quantity of digitized fine art collections. An applicable art-related image classification algorithm based on CNN fine-tuning experiments was suggested with higher accuracy outcomes (*Zhang, 2024*). The article researched the style identifications of oil painting images by using a deep learning method called ResNet-NTS with better outcomes when compared to similar algorithms (*Kelek, Calik & Yildirim, 2019*). The article investigated GoogleNet, DenseNet, ResNet50, ResNet101, and Inceptionv3

networks using the data set consisting of 17 painters with an average of 46 paintings per se who lived in distinct terms and were impacted by distinct art movements (*Leonarduzzi, Liu & Wang, 2018*). The article examined two databases of art objects (postimpressionist paintings and Renaissance drawings) to determine whether they were generated by van Gogh and Raphael, respectively, using a deep learning algorithm (*Castellano et al., 2022*). The manuscript proposed a novel KG-enabled fine art classification approach using *A r t G r a p*, which is implemented to conduct artwork feature estimations. A Knowledge Graph (KG) integrates a rich body of information about artworks, artists, and painting schools in a unified structured framework (*Liu et al., 2024*). More up-to-date research can be found in *Jin et al. (2024)*, *Guo, Yang & Ji (2024)* and *Chen & Shi (2021)*.

The research motivation of the article is to improve the classification performance of art images since the available art image classification methods have low recall and accuracy rate issues. Thus, a new adaptive classification method is designed employing multi-scale convolutional neural networks (CNNs). The proposed algorithm, called a multi-scale convolutional neural network (CNN), provides color recoveries by using enhancement processing of art images. Then, extreme pixel ratios and feature extraction are extracted to examine the quality of art images and use those scores as the initial values of the training.

The contribution of the article is summarized as follows:

(1) Using multi-scale Retinex algorithm to process images. Then, the extreme pixel ratio is used to evaluate the quality of artistic images and obtain high-quality artistic images for backup.

(2) The key features in art images are extracted by edge detection technology and used as the initial value of the item to be trained in the classification model.

(3) Based on the particularity of artistic image features, the neural network is constructed employing extended convolution.

(4) The decision fusion method based on maximum output probability is introduced to determine the final category of the input image according to the maximum output probability of different subclassifiers.

The rest of the manuscript is structured as follows: The enhanced processing methods of artistic images are provided in Section 'Enhanced processing of artistic images'. Section 'Feature extraction and analysis of artistic images' is allocated to feature extraction and its analysis. Section 'Construct a multi-scale convolutional neural network model' presents the proposed algorithm. Section 'Training of convolutional neural networks and processing of art image classification' is allocated to the training of the proposed algorithm. Experiments and their outcomes are presented in Section 'Experiment and Results'. Section 'Conclusion' concludes the research.

## ENHANCED PROCESSING OF ARTISTIC IMAGES

No universal image enhancement method exists that can be implemented for all types of images. Therefore, it is necessary to employ a specific class of processing methods to achieve better enhancement effects for images with different quality degradation, and the same is true for artistic images.

The decline in the quality of artistic images also manifests itself in many ways. One of the common reasons is that the original image has been treated with special artistic effects, and the creator wants to employ it to express a specific image (*Shen et al., 2023*). To meet this requirement, grayscale images, and histograms are selected to enhance original art images.

The gray statistical histogram of an image classifies the gray value of each pixel according to a certain gray level and finally presents the statistical results in the form of a column graph, reflecting the overall distribution of the pixel gray value of an image (*Han & Jie, 2024*; *Kim et al., 2023*). The histogram reflecting the distribution of pixel gray values can be utilized as an important statistical feature of an image. The histogram-based enhancement method remaps pixel gray values to a more ideal gray-scale range using the established rules.

When Single-scale Retinex (SSR) and Multi-Scale Retinex (MSR) are used to process color images, the conventional practice is to conduct a Retinex enhancement on the red-green-blue (RGB) three channels of color images, and then recombine the results of the three channels into a color image (*Kangkang, Qingliang & Songhao, 2024*). However, when the result is normalized to the displayable range in the last step of the algorithm, the dynamic range of the three channels may not be the same, so the mapping function used in the normalization is not the same, resulting in a serious color bias in the final image. Due to the stated shortcomings, the multi-scale Retinex algorithm with color restoration was used to complete the processing in this study. The Retinex method mainly consists of 2 steps: estimation and the normalization of illumination (*Aguirre-Castro et al., 2022*). Images captured under low light conditions often suffer from various degradations 35. The Retinex models are highly effective in enhancing low-light images. These algorithms perform the histogram equalization to distribute pixels, reduce the predominant color, perform color and contrast correction, and achieve an automatic white balance to improve illumination 36.

The first steps of the Multi-Scale Retinex Model with Color Restoration (MSRCR) algorithm are the same as those of the MSR. The difference comes from the last step of the normalization process, a color recovery factor $T_i(x, y)$ is introduced.

$$F_{MSRCR}(x, y) = T_i(x, y) \times F_{MSR}(x, y) \qquad (1)$$

$$T_i(x, y) = \gamma \left\{ \kappa \ln[X_i(x, y)] - \ln[X_j(x, y)] \right\} \qquad (2)$$

where $F_{MSRCR}(x, y)$ represents the result of the MSRCR correcting the color deviation, and the subscript $i$ indicates that the result corresponds to the $i$th channel of the color image, which is usually RGB three-channel. In Eq. (2), $\gamma$ represents the gain constant, $\kappa$ denotes the controlled nonlinear strength parameter, which needs to be adjusted according to the actual processing effect. where $X_i(x, y)$ represents the $i$th channel value of the original input image.

The MSRCR algorithm adjusts the proportional relationship between the three color channels in the original image by introducing a color recovery factor to eliminate color distortion (*Han & Jie, 2024*).

According to the ideas obtained from the analysis of Eqs. (1) and (2), the art image enhancement processing is given as follows:

Step 1: Input an art image to be processed.

Step 2: The RGB color space of the image is converted to the HSV color space, and the channel values are stored as floating point numbers.

Step 3: Traverse all pixels of the S-channel of the image and use Eq. (3) to perform a logarithmic transformation:

$$s' = \frac{\lg_2(as+1)}{\lg_2(a+1)} \tag{3}$$

where $s$ denotes the saturation score of each pixel in the input image channel, and $s'$ represents the saturation after logarithmic transformation; $a$ represents the parameter that controls the bending degree of the transformation curve.

Step 4: The HSV color space of the image is converted back to RGB color space, and the channel scores are delimited to integers within the specified interval by rounding the decimal number.

Step 5: Output the enhanced art image.

To ensure the quality of the image processing, the quality evaluation of the enhanced image is carried out. The literature points out that the color of art images with good quality is rarely distributed near the black-and-white ends of the brightness axis, but is relatively concentrated in the middle brightness range. Further analysis in this paper shows that the closer the color is to white, the lower its saturation, and the closer the color is to black, the lower its brightness (*Alzahem et al., 2023b*). Furthermore, an evaluation index proposed in this paper is the ratio of extreme pixels, and it is computed by Eqs. (4) and (5).

$$Z_1 = \sum_{i=1}^{n1} A_1 \tag{4}$$

$$Z_2 = \sum_{i=1}^{n2} A_2 \tag{5}$$

where $Z_1$ and $Z_2$ represent extreme pixel ratio, and subscripts 1 and 2 are used to distinguish between saturation and brightness, whose meaning is the total percentage of pixel points in a certain range on the left side of the horizontal axis in the histogram, $A_1$ and $A_2$ denote the column area of the $Z$ level from left to right on the horizontal axis of the saturation and brightness histogram, that is, the proportion of pixel points at this level, $n1$ and $n2$ represent the artificially specified extreme range widths in the saturation and brightness histograms, respectively, and can only be positive integers. The extreme range width can also be specified in the form of a ratio, as shown in Eq. (6), $d$ represents the range width in the form of a ratio, and $N$ designates the total number of grades divided by the horizontal axis of the histogram.

$$d = \frac{Z-1}{N-1}. \tag{6}$$

The larger the value of Eq. (6), the better the image enhancement effect. After image enhancement processing is completed, image feature extraction is constructed to obtain the basic features of artistic images, which provides a basis for subsequent research.

## FEATURE EXTRACTION AND ANALYSIS OF ARTISTIC IMAGES

"Edge" is the dividing line between those regions with different local features of an image manifested as the discontinuity of the local image, such as the mutations of the gray level and the texture structure. The edge detection of an image can help us find the local feature area with significant changes in image attributes, to reflect the important events and changes of attributes and the significant structural attributes of an image. The "line" is a pair of edges with the same image features in the intermediate region, that is, a pair of edges with a small distance to form a line (*Tang et al., 2023c*; *Jing et al., 2024*; *Shirabayashi, Braga & da Silva, 2023*).

A 64×64 pixel grayscale image is captured from the whole image, which not only contains the local stroke structure information of an image but also keeps the reasonable operation cost. In addition, local feature extraction should consider 2 factors:

(1) Detect the representative typical brush information that best reflects the painter's style;

(2) Locate the representative area that best represents the artistic characteristics of the whole image.

Considering factor 1, the article utilizes edge detection technology to extract key artistic features in art images. For factor 2, a window function is implemented to progressively scan all areas of the complete image to process all image trace information. The algorithm extracting local features is presented as follows:

Let $G(x, y)$ represent a grayscale image and extract all image trace information as follows:

$$R(x, y) = Y(\nabla_s(G(x, y)))$$ (7)

where $\nabla_s$ represents the Sobel operator, $Y(\cdot)$ represents a binary equation, which uses the sensitive threshold $\sigma$ to binarize the edge image of $G(x, y)$ after edge detection by Sobel operator to obtain a binary image $R(x, y)$. The pixels in $R(x, y)$ are defined as follows:

$$r(x, y) = \begin{cases} 1 \, if \, |\nabla_s(G(x, y))| > \sigma \\ 0 \, else \end{cases}.$$ (8)

After the calculation is completed, the local area is segmented. The fixed module segmentation method is simple and intuitive, and the experiment shows that the classification performance is very good. Inspired by image and video compression coding, the size of a local region can generally be defined as 128×128, 64×64, or 32×32. To obtain enough detailed features without forcing the algorithm to compute too much, this paper defines a 64×64 pixel image block through experiments to locate the local area in an image that best reflects the characteristics of the pen style:

$$block_k(i, j) = \eta(i, j)G(x - k\Delta, y - k\Delta)$$

**Peer**J Computer Science

$$\forall i \in [0,64], \forall j \in [0,64], \forall x \in [0,P-1], \forall y \in [0,Q-1] \tag{9}$$

where $0 \le k < N, N$ represents the total number of image blocks, $\Delta \in [0,64]$ represents the step size of the window function across the entire image.

To balance the operation speed and it needs to be chosen as many blocks as possible to include all the details, this paper chooses 1/4 of the width and height of the image block as the score of $\Delta$, that is, $\Delta = 16$. In Eq. (9), $\eta(i,j)$ represents a window function defined as follows:

$$\eta(i,j) = \begin{cases} 1 \forall i,j \in [0,64) \\ 0 \, else \end{cases}. \tag{10}$$

The image block denoted by $block_k(x,y)$ is the most representative artistic style that is extracted as follows:

$$block_k(x,y) = \text{argmax}\{\lambda block_k(R(x,y))\} \tag{11}$$

where $\lambda$ represents the counting function, which is defined as follows:

$$\lambda = \sum_{i=0}^{63} \sum_{j=0}^{63} r(i,j). \tag{12}$$

Therefore, $block_k(x,y)$ contains the most detailed artistic features and is the most representative local area extracted. Therefore, extracting the detailed features of artistic images from the local area block $block_k(x,y)$ containing the most strokes can best describe the creation style of artistic works.

## CONSTRUCT A MULTI-SCALE CONVOLUTIONAL NEURAL NETWORK MODEL

Based on the particular features of artistic images, the neural network is constructed employing extended convolution in this study.

In general, the receptive field plays a pivotal role in CNNs. The receptive field refers to the size of a region where an element on the feature map corresponds to the input map. In general, when using a CNN to resolve problems, the larger the receptive field, the better the network effect. The receptive field can be increased by increasing the number of layers in the network, increasing the size of the convolutional filter, and using the pooling layer (*Chen, Schoenhardt & GuMin, 2023*). Nowadays, extended convolution has been widely used to resolve various image problems. For example, extended convolution not only helps us resolve semantic segmentation and image classification problems but also helps resolve image regression problems such as image deblurring and image denoising (*Seema, Vasantha Lakshmi & Patvardhan, 2023*).

The expansion factor $d$ is an important parameter of the extended convolution. Assuming the convolution kernel size is $3 \times 3$, the receptive field $b_d$ of the network is calculated as follows:

$$b_d = (2^{d+1}-1) \times (2^{d+1}-1). \tag{13}$$

After the above setup, a neural network is established, which consists of seven DilBlock modules, six ResBlock modules, two convolutional nuclei of size 7×7, three convolutional nuclei of size 3×3, two upper sampling layers, and two convolutional layers.

The input image $a$ is divided into three levels after input, activation function RELU is $\partial(a) = \max(0, a)$, and normalization mode is denoted by $GN$. The first route is followed by a 1×1 convolution and a 3×3 convolution with two set expansion factors.

$$C_{11}(a) = \eta_{11}\partial(a) + h_{11} \tag{14}$$

where $\eta_{11}$ represents the convolution weight of layer 1, $h_{11}$ stands for layer 1 convolution bias.

$$C_{12}(a) = \eta_{12}\partial(C_{11}(a)) + h_{12}$$
$$C_{13}(a) = \eta_{13}\partial(C_{12}(a)) + h_{13} \tag{15}$$

where $\eta_{12}$ represents the convolution weight of layer 2, $h_{12}$ stands for layer 2 convolution bias, $\eta_{13}$ represents the convolution weight of the third layer, $h_{13}$ represents layer 3 convolution bias. The second route consists of a 1×1 convolution and a 3×3 convolution with a set expansion factor, giving the vector $C_{22}(a)$ as follows:

$$C_{21}(a) = \eta_{21}\partial(a) + h_{21}$$
$$C_{22}(a) = \eta_{22}\partial(C_{21}(a)) + h_{22} \tag{16}$$

where $\eta_{21}$ represents the convolution weight of layer 1, $h_{21}$ represents layer 1 convolution bias, C $\eta_{22}$ represents the convolution weight of layer 2, $h_{22}$ represents layer 2 convolution bias. Then, the output at this time is cascaded with the output of the first route to attain:

$$f_{concat}(a) = Concat\,[C_{13}(a), C_{22}(a)]. \tag{17}$$

Cascaded information is input into a 1×1 convolution to reduce and fuse information:

$$C_3(a) = \eta_3\partial(f_{concat}(a)) + h_3 \tag{18}$$

where $\eta_3$ and $h_3$ represent the weight and bias of the last layer of convolution respectively. Finally, to fuse the low-frequency and high-frequency information, a jump connection operation is performed with the input. After sorting out the above contents, the construction of the multi-scale convolutional neural network is completed. A multi-scale CNN can perform convolutional operations and attribute derivation on input data at multiple scales, capturing attribute information at distinct scales (*Yang & Liu, 2024*). The multi-scale network fusion strategy enhanced the classification precision of the model (*Yang et al., 2024*). The multi-scale transform can decompose the source image into components at distinct scales, and each component denotes a sub-image at each scale (*Tang et al., 2011*). Thus, the above three subnetworks employ the Softmax number for classification in the last fully connected layer, and the output of the Softmax number is a probability that the input sample belongs to each category.

## TRAINING OF CONVOLUTIONAL NEURAL NETWORKS AND PROCESSING OF ART IMAGE CLASSIFICATION

The decision fusion method based on maximum output probability is one of the simplest algorithms. The method calculates the output probability of different subclassifiers and

determines the final category of the input image according to the maximum score of the calculated results. If the maximum output probability of each classifier occurs in the same class, the decision result does not change. However, as the probability changes, the likelihood of the decision becomes very different. This method does not consider the probabilities as the weight of each classifier in decision fusion, but the weight of each classifier determines the classification effect of decision fusion.

Images of different sizes have distinct features extracted by multi-scale CNN, and samples have certain randomness, so it is necessary to formulate a decision method suitable for art images to obtain the optimal classification performance. In this study, the adaptive entropy-weighted decision fusion method is implemented to assign different fusion weights to distinct input art image features.

The adaptive entropy-weighted decision fusion algorithm is set as follows:

When the three Softmax functions of multi-scale CNN are output in parallel, the probability output matrix of each input art image can be obtained as follows:

$$L(a) = \begin{bmatrix} l_{11}(a)l_{12}(a)\dots & l_{1n}(a) \\ l_{21}(a)l_{22}(a)\dots & l_{2n}(a) \\ l_{31}(a)l_{32}(a)\dots & l_{3n}(a) \end{bmatrix}_{3n} \tag{19}$$

where each row represents the probability output value of Softmax number of 1 subnetwork against the input sample $a$, $n$ represents the number of art image types, and the column label with the highest probability of each row is the prediction category of Softmax number of each subnetwork against the sample.

The characteristics of probability scores belonging to each category of samples will affect the classification accuracy. The smaller the difference in probability values, the greater the uncertainty of classification. If the largest probability value is more different from the other probability values, the classification result is more reliable. Therefore, this paper introduces information entropy to represent the uncertainty of the Softmax number classification of input sample 2 by the second subnetwork, which is publicized as follows:

$$U_i(a) = -\sum_{j=1}^{n} l_{ij}(a)\lg l_{ij}(a) \tag{20}$$

where $l_{ij}(a)$ represents the probability that the Softmax number of the $i$th subnetwork will judge the input sample $a$ as belonging to the class $j$. If the information entropy value of the Softmax function is $a$, a subnetwork is larger, the classification uncertainty is higher, then the Softmax number of the network has poor classification ability for the input sample $a$, and the fusion weight of the Softmax □ number of the network is smaller, and vice versa. Therefore, the adaptive fusion weight calculation of the Softmax number of multi-scale CNN is presented as follows:

$$\varpi_i = \frac{\exp(-U_i(a))}{\sum_{l=1}^{3}\exp(-U_i(a))}. \tag{21}$$

After obtaining the fusion weights, each row of the probability output matrix $L(a)$ is multiplied by the weights to obtain a new probability output matrix $L'(a)$:

$$L'(a) = \begin{bmatrix} \varpi_1 l_{11}(a) \varpi_1 l_{12}(a) \dots \varpi_1 l_{1n}(a) \\ \varpi_2 l_{21}(a) \varpi_2 l_{22}(a) \dots \varpi_2 l_{2n}(a) \\ \varpi_3 l_{31}(a) \varpi_3 l_{32}(a) \dots \varpi_3 l_{3n}(a) \end{bmatrix}_{3n}. \tag{22}$$

$L'(a)$ is added to the weighted sum by column, then the label of the maximum value of the weighted sum is the result of decision fusion, as follows:

$$label(a) = \text{argmax} \left[ \sum_{l=1}^{3} \exp(\varpi_i l_{ij}(a)) \right]. \tag{23}$$

The adaptive entropy-weighted decision fusion algorithm fully considers the different classification effects of the Softmax function of different subnetworks on the same input art images and the different classification performance of Softmax number of the same subnetwork on different art images used as input, and adaptively assigns more reasonable fusion weights to different input art images. The misclassification problem of single-scale CNN generated by the output probability value of Softmax is resolved.

# EXPERIMENT AND RESULTS

## The preparation for the experiment

In the experiment, the Keras deep learning framework was adopted, Adam optimization was used, the learning rate was set to 0.001, and the training cycle was set to 120. The image samples in the training set were randomly rotated by 0°–15°, and randomly flipped and offset by 0%–5% in the horizontal and vertical directions, respectively, to enhance the generalization capability of the network. When the accuracy of network training is not improved after six training stages, the learning rate drops to less than 10% of the original score.

The experimental data was collected from the Artlib World Art Appreciation Database (https://www.artgalleria.com, Dayi.com (http://www.dayi.com), and other websites, including 3,393 prints, 4,500 Chinese paintings, 1,350 oil paintings, 3,300 gouache paintings, and 3,415 watercolors. The style distribution information of all kinds of art images was relatively uniform. The images with high resolution and rich style information are cut into several 299×299-pixel images, and the images with rich style information are manually screened to enhance the data. After data enhancement, 5,000 prints, 5,141 Chinese paintings, 2,401 oil paintings, 3,300 watercolors, and 4,201 gouache paintings were randomly selected for the experiment.

To facilitate experimental analysis, the following images were selected as important experimental targets in this study to complete the experimental process.

In the experiment, recall rate, accuracy rate, and ROC curve were used as indexes to assess the performance.

Suppose there are $N$ paintings to be identified in the experimental image library, and after classification, $Q_1$ denotes the number of paintings of this type correctly classified, and

**Table 1 Experimental results of classification recall rate and precision rate (unit: %).**

| Type | Method of this paper | | Method of reference (*Zhao et al., 2022*) | | Method of reference (*Su, Wang & Zhang, 2023*) | | Method of reference (*Tang et al., 2023a*) | |
|---|---|---|---|---|---|---|---|---|
| | Recall factor | Precision rate | Recall factor | Precision rate | Recall factor | Precision rate | Recall factor | Precision rate |
| Print | 95.8 | 85.6 | 92.1 | 84.1 | 90.2 | 80.5 | 94.5 | 79.5 |
| Traditional Chinese painting | 96.2 | 85.3 | 90.1 | 82.0 | 91.0 | 80.1 | 93.2 | 75.5 |
| Oil painting | 95.0 | 85.0 | 92.0 | 81.5 | 91.5 | 82.1 | 91.5 | 74.6 |
| Painting gouache | 94.6 | 85.0 | 93.5 | 80.1 | 92.3 | 80.0 | 92.0 | 78.1 |
| Watercolour painting | 94.5 | 85.0 | 93.2 | 75.0 | 93.0 | 75.5 | 91.0 | 76.5 |

$Q_2$ represents the number of paintings of this type incorrectly classified, then the accuracy rate $\delta$ andrecall rate $\mu$ are defined by Eqs. (24) and (25)

$$\delta = \frac{Q_1}{Q_1 + Q_2} \tag{24}$$

$$\mu = \frac{Q_1}{N}. \tag{25}$$

After setting experimental indexes, the methods in *Zhao et al. (2022)*, *Su, Wang & Zhang (2023)* and *Tang et al. (2023a)* were selected to compare the proposed method.

## The analysis of the results
### Experimental results of recall and precision
Combining Eqs. (24) and (25), the classification results of various types of images are analyzed. A comparative analysis using the four different classification methods is obtained for the test set composed of five types of art images, as shown in Table 1.

Table 1 depicts the experimental results of the proposed method in the five types of art images that are superior to the three comparison methods, indicating that the proposed can more effectively describe the artistic styles of different painting methods, to identify art categories more effectively. It provides remarkable results. Its accuracy is not lower than 85.0%, and the maximum recall rate can reach 96.2%, indicating that the proposed method can be practically implemented according to distinct styles with high robustness.

## Experimental results of the ROC curve
More robust outcomes are presented with ROC curves. The larger the area value of the ROC curve, the better the classification performance. ROC curves of the three methods are shown in Fig. 1.

(a) The proposed method
(b) The method in *Zhao et al. (2022)*
(c) The method in *Su, Wang & Zhang (2023)*
(d) The method in *Tang et al. (2023a)*

Figure 1 depicts the probability of each algorithm. Each figure presents the results of each method used in the analysis, namely the methods in *Zhao et al. (2022)*, *Su, Wang &*

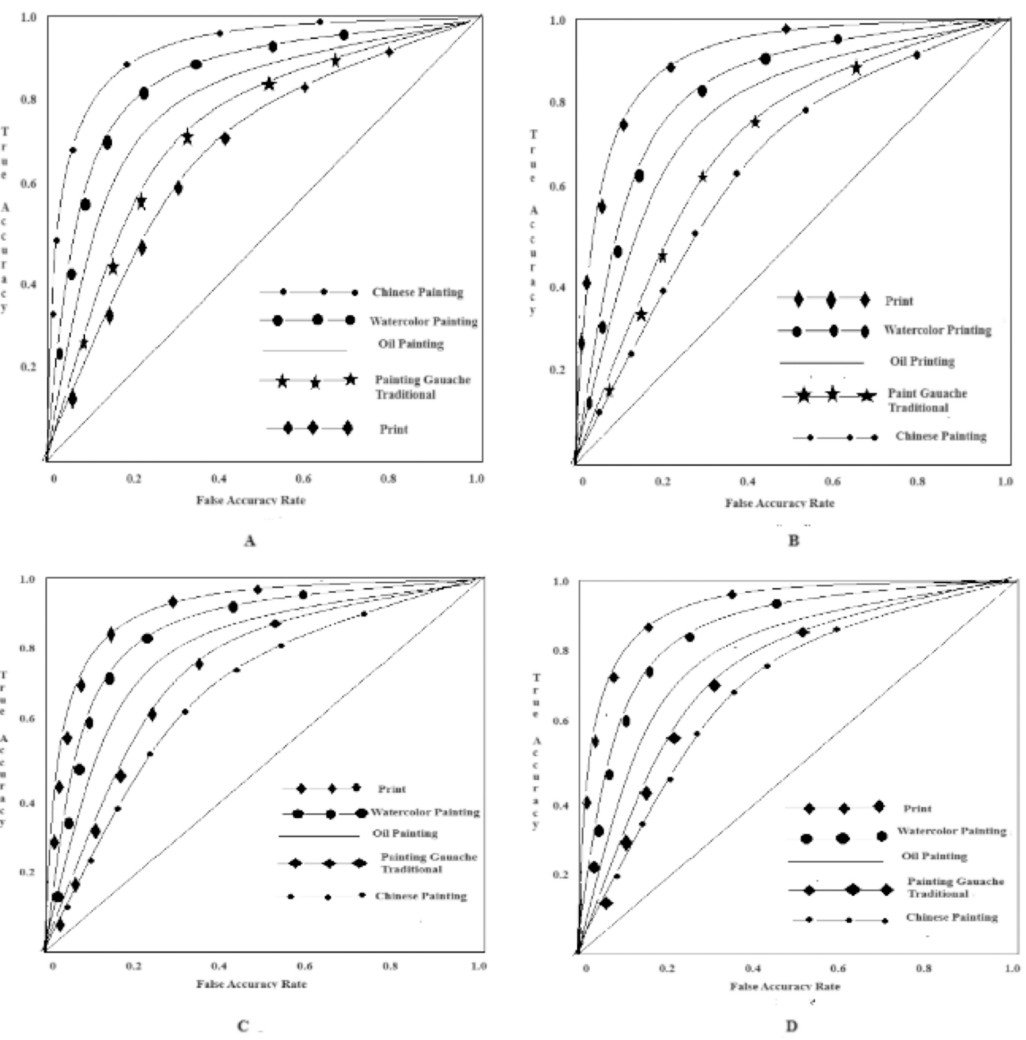

**Figure 1** Test results of the ROC curve.

*Zhang (2023)* and *Tang et al. (2023a)* and the proposed method. Even though the other methods provide high classification accuracies, the proposed method excels them.

The results show that when the print score reaches 0.99, the classifier has the best classification result for prints, followed by Chinese paintings. Because the color blocks, lines, painting techniques, and other features of oil painting are similar to other kinds of painting in overall characteristics and local details, the ROC curve of oil painting has the smallest range. The ROC curve of the proposed method is stable and the numerical score is high. It can be confirmed that the proposed method has a high and stable classification performance, and can be applied to the classification of artistic images in practical implementations.

## CONCLUSION

The research work of art image management mainly focuses on the classification of paintings according to the subject matter and expression technique, the classification of painters according to the creation style, and the identification of true and false art portraits. The proposed method contributes to the area of insufficient research when the classification of many kinds of art images is required. The proposed method attains a better classification effect on art images than the available network model and conventional classification method. In the subsequent work, we will optimize the art image classification network model, expand the art image sample base, and further improve the accuracy and efficiency of the network classification model.

### Funding
This study was suppored by Research on the Design of Integrated Material Creation Based on Fractal Theory, Research Funding of Wuhan Polytechnic University (No. 2022RZ079). The funders had no role in study design, data collection and analysis, decision to publish, or preparation of the manuscript.

### Grant Disclosures
The following grant information was disclosed by the authors:
Research on the Design of Integrated Material Creation Based on Fractal Theory, Research Funding of Wuhan Polytechnic University: No. 2022RZ079.

### Competing Interests
The authors declare there are no competing interests.

### Author Contributions
- Jin Xiang conceived and designed the experiments, performed the experiments, analyzed the data, prepared figures and/or tables, and approved the final draft.
- Yi Yang conceived and designed the experiments, performed the experiments, analyzed the data, prepared figures and/or tables, authored or reviewed drafts of the article, and approved the final draft.
- Junwei Bai conceived and designed the experiments, performed the experiments, performed the computation work, prepared figures and/or tables, and approved the final draft.

### Data Availability
This work evaluated images from Pixabay.

### Supplemental Information
Supplemental information for this article can be found online at http://dx.doi.org/10.7717/peerj-cs.2336#supplemental-information.

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
