# Peer review of "Adaptive classification of artistic images using multi-scale convolutional neural networks"

_PeerJ Computer Science, doi:10.7717/peerj-cs.2336_

## Round 0.1 · original submission · Major Revisions

Dear authors,

Thank you for submitting your article. Based on this reviewers' comments, your article has not yet been recommended for publication in its current form. However, we encourage you to address the concerns and criticisms of the reviewer and to resubmit your article once you have updated it accordingly. Reviewer 2 has asked you to provide specific references. You are welcome to add them if you think they are relevant. However, you are not obliged to include these citations, and if you do not, it will not affect my decision.

Best wishes,

Reviewer 1 ·

Basic reporting

- The introduction part of the study is quite weak. Since there is no literature review section in the paper, more studies on the problem should be included in the introduction section. Then, based on the shortcomings in existing studies, motivation had to be clearly stated. But motivation is also lacking. Not enough information is given in the introduction section about the problem being studied. The contributions are not well reviewed in the introduction. For example; It has been stated that a multi-scale Retinex algorithm was used to process the images. It should be more clearly stated what purpose this algorithm is used in image processing. Other contributions should also have been examined from this perspective.
- The article contains grammatical errors.
- Readers expect the organization of the paper at the end of the introduction.
- What are SSR, MSR, MSRCR abbreviations? SSR appears only once throughout the article. What is SSR?
- The flowchart or pseudocode of the proposed method should be included in the article.
- The resolution of the images is quite low. The texts in the images cannot be read.
- The web pages from which the data sets are taken should be cited in accordance with the reference style of the journal, rather than stating them in the text.

Experimental design

- The experimental results obtained with the proposed method are quite limited. I believe that more and more comprehensive experiments should be conducted in articles written in SCI journals.
- Comments on the experimental results are few. For example, Figure 3 was not evaluated.

Validity of the findings

No comment

Additional comments

I carefully examined the article titled "Adaptive classification of artistic images based on multi-scale convolutional neural networks". There are major shortcomings in the article.

Reviewer 2 ·

Basic reporting

This manuscript to improve the classification performance of art images, presented a new adaptive classification method is designed using multi-scale convolutional neural networks. Overall, the structure of this paper is well organized, and the presentation is clear. However, there are still some crucial problems that need to be carefully addressed before a possible publication. More specifically,

1.More adequate motivation and justification for the use of multi-scale convolutional neural networks is needed, otherwise it is not clear to me what obvious value the manuscript brings to the community.
2. Writing of Section 2, 3, 4 and 5. looks like a software manual, there is no intuition behind each step.
3. Why use Retinex algorithm? Is Retinex the best choice? This requires full research and a full explanation, I suggest the author add more ablation studies here.
4. The reviewer is wondering how about the computational complexity of the proposed method?

5. A deep literature reviews should be given, particularly advanced and SOTA Deep learning algorithms, or feature fusion models. Therefore, the reviewer suggests cite the following works in the revised manuscript, such as:

(1) Liu, J., Chen, Z., Zhou, J., Xue, A., Peng, D., Gu, Y., & Chen, H. (2024). Research on A Ship Trajectory Classification Method Based on Deep Learning. Chinese Journal of Information Fusion, 1(1), 3-15. (2) Jin, X., Tong, A., Ge, X., Ma, H., Li, J., Fu, H., & Gao, L. (2024). YOLOv7-Bw: A Dense Small Object Efficient Detector Based on Remote Sensing Image. IECE Transactions on Intelligent Systematics, 1(1), 30-39. (3) Guo, X., Yang, F., & Ji, L. (2024). A Mimic Fusion Algorithm for Dual Channel Video Based on Possibility Distribution Synthesis Theory. Chinese Journal of Information Fusion, 1(1), 33-49.

6. In addition, the manuscript should have a section to describe state-of-the-art techniques. This section should also outline a tabular sketch so that it is easy to identify what's missing in the literature and how this paper addresses that. This section can be derived from contents described in the introduction section.
7. The experiments in the manuscript are difficult to demonstrate the superiority of the proposed method, especially Retinex and multi-scale CNN also lack convincing ablation experiments.

Experimental design

The experiments in the manuscript are difficult to demonstrate the superiority of the proposed method, especially Retinex and multi-scale CNN also lack convincing ablation experiments.

Validity of the findings

The proposed technique exhibits effective results.

Additional comments

N/A

Reviewer 3 ·

Basic reporting

Proofread the text to improve grammar and readability. For example, "Finally, a multi-scale convolutional neural network is constructed by using extended convolutions" could be rephrased for clarity.

Experimental design

Explain why art image classification is particularly challenging and why it necessitates a new approach.
• Please break the methodology into sub-sections such as Image Enhancement, Feature Extraction, and Classification Model.
• The use of the multi-scale Retinex algorithm with color recovery is mentioned briefly. Provide a short explanation of what this algorithm does and why it is suitable for art image enhancement.
• The use of "extreme pixel ratio" to evaluate image quality is mentioned but not explained. Clarify what this ratio is, how it is calculated, and why it is a good measure of image quality for art images.
• Explain how the key features extracted from edge detection are used as initial values for the classification model. Provide more detail on the process of integrating these features into the model.
• The construction of the multi-scale convolutional neural network (CNN) with extended convolutions is mentioned. Explain what extended convolutions are and how they differ from standard convolutions.
• Describe the architecture of the multi-scale CNN. How are the different scales integrated, and what are the characteristics of each level network?

Validity of the findings

• The results section should provide more quantitative data. Include metrics such as recall, precision, F1-score, and any relevant comparisons with baseline methods.
• Provide visual examples or case studies that illustrate the effectiveness of the proposed method

---

## Round 0.2 · accepted · Accept

Dear authors,

Thank you for the revision and for clearly addressing the reviewers' comments. Although Reviewer 1 writes that his/her suggested revisions were not carried out and does not agree with the acceptance of the paper, there is no explanation of which aspects of his/her review were not properly addressed. I can see that his/her concerns are addressed and confirm that the paper is improved. The paper seems now acceptable for publication in light of this revision.

Best wishes,

Reviewer 1 ·

Basic reporting

In the previous revision, my decision was to reject the paper. Also, the suggested revisions were not made, so my decision is still to reject the paper.

Experimental design

N/A

Validity of the findings

N/A

Additional comments

N/A

Reviewer 2 ·

Basic reporting

The authors have dealt with the points I raised up.

Experimental design

No further comments

Validity of the findings

No further comments

Additional comments

N/A

Reviewer 3 ·

Basic reporting

The authors have solved correctly all the issues I pointed out in the first round of the review.

Experimental design

The authors have solved correctly all the issues I pointed out in the first round of the review.

Validity of the findings

The authors have solved correctly all the issues I pointed out in the first round of the review.